# Attenuation of Ventilation-Induced Endoplasmic Reticulum Stress Associated with Lung Injury Through Phosphoinositide 3-Kinase-Gamma in a Murine Endotoxemia Model

**DOI:** 10.3390/ijms26125761

**Published:** 2025-06-16

**Authors:** Li-Fu Li, Chung-Chieh Yu, Chih-Yu Huang, Huang-Pin Wu, Chien-Ming Chu, Ping-Chi Liu, Yung-Yang Liu

**Affiliations:** 1Division of Pulmonary and Critical Care Medicine, Department of Internal Medicine, Chang Gung Memorial Hospital, Keelung 20401, Taiwan; lfp3434@cgmh.org.tw (L.-F.L.);; 2Department of Internal Medicine, Chang Gung University, Taoyuan 33302, Taiwan; 3Department of Respiratory Therapy, Chang Gung Memorial Hospital, Keelung 20401, Taiwan; 4Community Medicine Research Center, Chang Gung Memorial Hospital, Keelung 20401, Taiwan; 5Chest Department, Taipei Veterans General Hospital, Taipei 112201, Taiwan; 6School of Medicine, Faculty of Medicine, National Yang Ming Chiao Tung University, Taipei 112304, Taiwan; 7Institute of Clinical Medicine, School of Medicine, National Yang Ming Chiao Tung University, Taipei 112304, Taiwan

**Keywords:** endoplasmic reticulum stress, phosphoinositide 3-kinase-γ, sepsis, ventilator-induced lung injury

## Abstract

Patients with sepsis often receive mechanical ventilation (MV). Continued use of MV may increase overdistention in the lungs, inflammatory mediator production, and inflammatory cell recruitment, eventually causing ventilator-induced lung injury (VILI). Endoplasmic reticulum (ER) stress caused by MV, oxidative stress, and sepsis results in dissociation of GRP78 from transmembrane proteins (PERK, IRE1α, and ATF6) and generates abundant incorrect protein structures. Phosphoinositide 3-kinase-γ (PI3K-γ) has been demonstrated to modulate ER stress associated with sepsis and acute lung injury (ALI). However, the regulatory mechanisms by which ER stress is involved in VILI remain unclear. In this study, MV was hypothesized to augment lung injury and induce ER stress through the PI3K-γ pathway, regardless of endotoxemia. Wild-type or PI3K-γ-deficient C57BL/6 mice were exposed to 30 mL/kg tidal volume of MV with or without endotoxemia for 5 h. The control group comprised nonventilated mice. MV with endotoxemia increased microvascular permeability, lung edema, interleukin-6 and metalloproteinase-9 production, oxidative loads, ER stress biomarkers (GRP78, IRE-1α, PERK), morphological rearrangement, PI3K-γ expression, and bronchial epithelial apoptosis in rodent lungs. The increase in lung injury was substantially reduced in PI3K-γ-deficient mice and in mice administered 4-phenylbutyric acid. In conclusion, MV-augmented ALI after endotoxemia partially depends on the PI3K-γ pathway.

## 1. Introduction

Sepsis is a primary reason for admission to the intensive care unit. Sepsis causes profound systemic inflammation and has a poor prognosis [1,2,3]. Lipopolysaccharides (LPSs) induce inflammatory cascades leading to the production of cytokines interleukin (IL)-6 and metalloproteinase-9 (MMP-9) [1,2,3]. Cytokine production can cause alveolar and vascular endothelial cell injury, hypoxemia, lung edema, microvascular leakage, and multiple organ system failure [1,2,3]. Mechanical ventilation (MV) is used as life support for patients with sepsis-associated acute lung injury (ALI). Continued use of MV may increase the risk of pathologic lung overinflation, increase inflammation, increase the production of inflammatory mediators, and cause ventilator-induced lung injury (VILI) [4]. Aberrant mechanical stretching can cause cellular barrier dysfunction, cytotoxicity, inflammation, and metabolic dysfunction [5,6]. MV induces oxidative load and endoplasmic reticulum (ER) stress in the airway epithelial cells, which deliver injury-related molecules with damage-associated molecular patterns that cause severe lung inflammation [5,6]. The effects of sepsis and MV on ER stress and lung injury warrant further investigation.

ER is an essential biosynthetic component that plays a major role in maintaining lipid, protein, and redox homeostasis and in controlling secretory processes [7]. ER stress and its signaling network, unfolded protein response (UPR), function as a key adaptation, control, and sensing mechanism between intercellular signaling pathways, enabling cell adaptation to various metabolic, oxidative/reductive stress, and environmental alterations [7]. ER stress influences three transmembrane sensors: activating transcription factor 6 (ATF6), inositol-requiring enzyme 1α (IRE1α), and protein kinase RNA-like endoplasmic reticulum kinase (PERK) [7,8,9]. As an endogenous chaperone in the ER, 78 kDa glucose-regulated protein (GRP78) can modulate the activation of ER stress sensors [10,11]. ER stress induced by pathological conditions, such as MV, oxidative stress, and sepsis, causes GRP78 to dissociate from transmembrane proteins, which are then activated to sense ER stress and initiate the UPR signaling [6,10,11]. GRP78 stimulates protein modification, oligomerization, and refolding in the ER for structural correction. However, the protein refolding produces reactive oxygen species (ROS) that aggravate oxidative stress. GRP78 expression functions as an indicator of ER stress [9].

IRE1 is the primary molecular branch of the UPR and comprises an endonuclease and a kinase domain [8,10]. When stimulated, IRE1α undergoes homo-oligomerization and transautophosphorylation, activating endoribonuclease and subsequently cutting a 26-nucleotide intron out of the mRNA coding for X-box binding protein (XBP-1) [10,11]. This process leads to the translation of spliced XBP-1, which translocates to the nucleus to increase the production of folding chaperones and other ER-resident proteins involved in lipid biosynthesis and ER-associated protein degradation [5,10,11]. PERK has been observed to inhibit translation through eukaryotic translation initiation factor 2α and induce transcriptional responses through activation of ATF4 and CCAAT/enhancer-binding protein homologous protein (CHOP) [12]. ATF6 may promote the production of proteins that augment ER function [5]. A study demonstrated an association between ER stress and VILI [10]. The mechanisms through which ER stress regulates VILI remain unclear, indicating a need for further investigation.

ER stress is involved in apoptosis regulation across numerous diseases [9]. IRE1 and PERK modulate apoptosis induced by ER stress in T cells during sepsis [11]. ER stress inhibition may ameliorate pathological organ injury. ER stress exacerbates cell death in the form of intrinsic apoptosis [13]. CHOP, a 29 kDa protein, plays a dual role in apoptosis induced by ER stress as both an inhibitor of C/EBPs function and an activator of other genes [12]. Under nonstress conditions, CHOP exists in the cytoplasm. Stress activates CHOP, causing it to aggregate in the nucleus. Extensive ER stress considerably increases CHOP expression [12,13]. Additionally, when stimulated by ER stress, caspase-12 enhances the downstream cell death molecule caspase-3 [12]. The upstream signaling pathways involved in ER stress regulation remain unclear.

Phosphoinositide 3-kinase (PI3K) is a cellular lipid kinase that phosphorylates the 3-hydroxyl of the phosphatidylinositol ring to generate the lipid second messenger phosphatidylinositol 3,4,5-triphosphate [14]. Studies that examine class I PI3Ks, comprising class IA (p110α, β, and δ) and IB (p110γ) isoforms, have been conducted [6,7]. Class IA kinases that form complexes with regulatory p85-related subunits containing SH2 are typically activated through receptor tyrosine kinases. Class IB p110γ binds with G protein–coupled receptors through its regulatory subunit p101 or p84 and G protein subunits [14]. PI3Ks and serine/threonine kinase/protein kinase B (Akt) pathways coordinate to regulate ER stress and LPS-induced ALI [13].

This study established a LPS-induced VILI model in mice and had the following objectives: (1) examine the association between PI3K-γ expression and the development of lung injury during MV; (2) compare oxidative load, inflammatory cytokine response, and ER response markers between MV and endotoxin-augmented lung injury; (3) investigate the role of PI3K-γ signaling in sepsis-induced ER stress through PI3K-γ homozygous knockout and pharmacologic inhibition using 4-phenylbutyric acid (4-PBA), an ER stress inhibitor [15]; and (4) clarify the role of PI3K-γ signaling in bronchial epithelia apoptosis due to sepsis. We hypothesized that MV damages the lungs, promotes free radical formation, and induces ER stress, regardless of LPS administration. We also hypothesized that these effects are elicited through the PI3K-γ pathway.

## 2. Results

### 2.1. Suppression of Endotoxin-Enhanced MV-Induced Microvascular Leak, Lung Edema, Hypoxemia, ER Accumulation, Oxidative Stress, IL-6, and MMP-9 Production, and ER Stress Protein Expression Through 4-PBA

ER stress and VILI in mice were induced using MV (tidal volume [V_T_] = 30 mL/kg) with room air for 5 h. We examined the injurious effects of overdistension and the treatment effects of intraperitoneally delivered 4-PBA, an ER stress inhibitor [15]. Physiological conditions upon the initiation and conclusion of MV are presented in Appendix A. Mice were kept in a euvolemic state through measurement of mean arterial pressure. The injurious effects of MV-induced morphological changes in the ER, microvascular permeability, lung water content, and hypoxemia were identified based on transmission electron microscopy (TEM) measurements (Figure 1A–E), lung endothelial barrier dysfunction (Figure 1F), lung total protein (Figure 1G), and gas exchange (partial pressure of oxygen [PaO_2_]/fraction of inspired oxygen [FiO_2_], Figure 1H). Inflammatory cytokine levels and oxidant load were measured to assess oxidative stress and determine the amount of inflammatory cytokines in the bronchial epithelium of mice with VILI (Figure 2A–D). The ER in mice with endotoxemia subjected to V_T_ = 30 mL/kg was markedly dilated and exhibited morphological changes, including a series of convoluted, flattened membrane sheets (cisternae) adjacent to the mitochondria (Figure 1A–E). These mice also exhibited higher endothelial barrier dysfunction levels, total protein, malondialdehyde (an aldehydic secondary product of lipid peroxidation), IL-6, and MMP-9 protein production and lower sodium dismutase production than mice without endotoxemia subjected to V_T_ = 30 mL/kg and the nonventilated control mice (Figure 1 and Figure 2A–D). The deterioration of lung injury in mice with endotoxemia and those subjected to high tidal volume MV substantially improved after 4-PBA administration (Figure 1 and Figure 2A–D). Western blot analyses revealed greater GRP78, phopho-IRE1α, phosphor-PERK, and cleaved ATF-6 expression in mice with endotoxemia subjected to V_T_ = 30 mL/kg than in those without endotoxemia subjected to V_T_ = 30 mL/kg and the nonventilated control mice (Figure 2E–H). The increased GRP78, phospho-IRE1α, phospho-PERK, and cleaved ATF-6 expression induced by endotoxemia and MV at V_T_ = 30 mL/kg were substantially reduced after administration of ER stress inhibitor 4-PBA (Figure 2E–H).

### 2.2. Endotoxin-Augmented MV-Induced PI3K-γ Protein Expression

PI3K activation has been demonstrated to modulate endotoxin-mediated ER stress [13,16]. Accordingly, we measured PI3K-γ expression to explore the role of the PI3K-γ pathway in VILI (Figure 3A). Western blot analyses revealed increased PI3K-γ expression in mice with endotoxemia subjected to V_T_ = 30 mL/kg than in those without endotoxemia subjected to V_T_ = 30 mL/kg and the nonventilated control mice. In mice with endotoxemia subjected to V_T_ = 30 mL/kg, 4-PBA prevented the activation of GRP78, phospho-IRE1α, phospho-PERK, and ATF-6 induced by MV with V_T_ = 30 mL/kg (Figure 2E–H). No decrease in PI3K-γ expression after 4-PBA administration was observed, indicating that ER stress markers are downstream in the PI3K-γ-induced signaling cascade. The cell types involved in the lung inflammation caused by mechanical stretching and the effects of 4-PBA on PI3K-γ activation in mice with VILI were determined through immunohistochemistry (Figure 3B,C). In positive immunohistochemical staining results for PI3K-γ, substantially higher levels were detected in the airway epithelial cells of mice treated with endotoxin and subjected to V_T_ = 30 mL/kg MV than in those without endotoxemia subjected to V_T_ = 30 mL/kg, and nonventilated control mice (Figure 3B,C). Consistent with the results of Western blot analyses, 4-PBA administration did not reduce the activation of PI3K-γ induced by high tidal volume MV (Figure 3B,C). Further experiments in PI3K-γ-deficient mice are warranted to elucidate the relationship between PI3K-γ and ER stress markers.

### 2.3. Reduction In Endotoxin-Augmented MV-Induced Lung Inflammation and ER Stress in PI3K-γ-Deficient Mice

PI3K-γ-deficient mice were utilized to determine the role of PI3K-γ in lung injury induced by stretching resulting from MV. In particular, we investigated whether the improvements in lung injury and ER stress after 4-PBA administration were achieved through PI3K-γ expression (Figure 4 and Figure 5). The alterations of MV on changes in microvascular permeability and pulmonary edema, hypoxemia, inflammatory cytokine generation, oxidative stress, and ER stress markers were significantly lower in PI3K-γ-deficient mice treated with endotoxin and high tidal volume MV (*p* < 0.05; Figure 4 and Figure 5).

### 2.4. Inhibition of MV-Induced Effects on Endotoxin-Enhanced CHOP Expression, Caspase-3, and Epithelial Apoptosis in PI3K-γ-Deficient Mice

In addition to contributing to oxidative stress, CHOP and caspase-3 play a key role in the intrinsic apoptotic pathway [12,17]. CHOP and caspase-3 expression were measured, and TEM was used to assess the functions of CHOP and caspase-3 pathways and the apoptosis of airway epithelial cells in endotoxin-associated VILI (Figure 6). In comparison to mice without endotoxemia subjected to V_T_ = 30 mL/kg and nonventilated control mice, a substantial increase in CHOP and cleaved caspase-3 (active form) expression was observed in the mice treated with endotoxin and high tidal volume MV (Figure 6). Epithelial apoptosis was represented by nuclear condensation and TUNEL-positive apoptotic nuclei of the bronchial epithelium in the mice treated with MV and endotoxin (Figure 6C,D and Appendix A). CHOP and caspase-3 activity and apoptosis in the airway epithelia stimulated by treatment with MV and endotoxin were lower in PI3K-γ-deficient mice than in control mice (*p* < 0.05; Figure 6). No significant differences were detected between wild-type mice and PI3k-γ-deficient mice in the nonventilated control group or between wild-type mice and 4-PBA-treated mice in the nonventilated control group, regardless of endotoxin administration; thus, these data are not presented herein. Our results indicate that inhibition of the PI3K-γ pathway suppressed MV- and endotoxin-induced inflammatory processes and ER stress (Figure 7).

## 3. Discussion

Sepsis is a severe disease that causes overwhelming systemic inflammation and subsequent multiple organ failure due to a dysregulated host response to infection. IL-6 and MMP-9 expression activate host defense mechanisms, which have been linked to increased inflammation and vascular permeability and serve as sepsis biomarkers [18,19]. Sepsis is a major cause of death worldwide, with a high mortality rate (25% to 42.3%) [20]. Patients with sepsis often develop acute respiratory failure, necessitating MV to improve oxygenation and ventilation, which can lead to the development of VILI [21]. Increasing evidence suggests that MV with even moderate tidal volumes can exacerbate sepsis-induced lung injury through inflammatory cell recruitment and upregulation of pulmonary cytokines, instigating ALI progression [22]. To resolve this clinical problem, recognizing pathogenic mechanisms and developing novel therapeutic agents is crucial. In this study, we applied our previously documented animal model to simulate VILI in endotoxemic mice to investigate the role of PI3K-γ in MV-induced ER stress and associated lung injury [23]. This study is the first to demonstrate markedly elevated PI3K-γ expression and ER stress following sepsis-induced ALI with VILI. The knockout of PI3K-γ or administration of the ER stress inhibitor 4-PBA was observed to (1) decrease oxidative stress and increase antioxidant activity; (2) mitigate inflammatory cytokines IL-6 and MMP-9; (3) attenuate the MV-induced increase in vascular permeability and lung edema; (4) reduce overexpression of the ER stress response protein GRP78 and ER transmembrane sensors PERK, IRE1α, and ATF6; (5) reduce CHOP, caspase-3 activity, and epithelial apoptosis; (6) improve the integrity of the ultrastructural ER tubules and mitochondria; and (7) restore functional gas exchange through the PaO2/FiO2 index. Additionally, we examined the role of PI3K-γ in mediating the deleterious effects of ER stress in our animal model. Notably, PI3K-γ overexpression was induced in the VILI group and was most severe in the group with VILI and LPS. ER stress markers were attenuated through knockout of PI3K-γ; nevertheless, inhibition of ER stress through 4-PBA administration did not significantly suppress PI3K-γ expression. This finding may imply that lung inflammation, apoptosis, and lung injury associated with ER stress were mediated through the PI3K-γ signaling pathway.

The ER, a vital intracellular organelle, facilitates protein folding, translocation, and posttranslational modifications in the secretory pathways [24]. Under inflammatory conditions, ER homeostasis is disturbed, indicating ER stress. During ER stress, UPR is produced to recover ER function. When ER stress overwhelms the UPR, it triggers apoptosis, causing cellular injury and death [24]. Reactive oxygen species, including free radicals, are produced by the oxidative folding system in the ER, activated by UPR. Oxidative stress negatively influences ER and protein homeostasis, resulting in mitochondrial damage [5,25,26]. Growing evidence has demonstrated that ER stress participates in sepsis and VILI pathogenesis [27,28]. In cecal ligation and puncture models of sepsis, ER stress contributed to abnormal lymphocyte apoptosis, and apoptotic pathways mediated by ER stress may have contributed to sepsis progression in mice [29]. A study indicated that the ER stress pathway and UPR-related markers involving CHOP play a pivotal role in responses to the pathogenesis of LPS-induced inflammation and apoptosis [30]. Inhibition of ER stress by 4-PBA has been observed to suppress LPS-induced lung inflammation and apoptotic reactions in the bronchial epithelial cells of both mice and humans [27,31]. ER stress may be more severe in clinical patients with sepsis who experience worse outcomes than in those experiencing favorable outcomes [32]. Patients with sepsis often require MV support for acute respiratory failure. Increasing studies have demonstrated that stretching due to MV induces ER stress and integrated stress response UPR in VILI models of alveolar epithelial cell stretch in rodents and other animals [8,10,11,15,28,33]. Protracted and aberrant unfolded or misfolded protein load in the ER activates three cytosolic sensors: PERK, IRE1α, and ATF6, each of which possesses different sensitivities to various ER stress patterns [34,35]. Dolinay et al. demonstrated that PERK serves as the principal activator of the ER response to mechanical stretch and that PERK inhibition decreases inflammatory signaling and restores function to the alveolar epithelial barrier, consequently improving VILI [28]. Moreover, Ye et al. showed that high V_T_ ventilation activated GRP78, CHOP, and IRE1α expression [10]; as well as VILI can induce ER stress and mitochondrial dysfunction [11]. In this study, we demonstrated that sepsis-induced ALI augmented by MV upregulated the major ER chaperone and stress response protein GRP78 by releasing from three UPR sensors. ER stress activated PERK, IRE1α, and ATF6, and the downstream signaling CHOP, caspase-3, and apoptotic pathways. TEM images of ultrastructural alterations in the ER displayed aberrant and enlarged tubular structures in the alveolar epithelial cells of mice receiving high V_T_ and more serious ultrastructural changes, such as the predominance of stripped ribosomes and destroyed mitochondria, in mice with sepsis receiving high V_T_. These deleterious effects and the consequent impaired oxygenation index (PaO_2_/FiO_2_ ratio) were alleviated by ER stress inhibitor 4-PBA, implying that ER stress may play a key mediating role in the exacerbation of sepsis-induced ALI by MV treatment.

The PI3K family comprises isoforms of heterodimeric lipid-modifying proteins that regulate numerous cellular functions, including cell growth, proliferation, migration, adhesion, and survival [36]. PI3K-γ is predominantly expressed in immune cells, where it modulates leukocyte activation and migration into injured tissues, and has been established as a critical regulator of inflammatory and immune responses [36]. In our previous study, epithelial–mesenchymal transition that emerged following bleomycin-induced ALI was aggravated by treatment with MV through PI3K-γ, indicating PI3K-γ plays a pivotal role in mediating pulmonary fibrogenesis after ALI [37]. PI3K has also been established as an upstream activator of ER stress, involving lung fibroblast proliferation, causing bleomycin-induced lung fibrosis [38]. Notably, PI3K-γ signaling is associated with leukocyte-dependent inflammatory responses in sepsis and VILI [38,39,40,41]. PI3K-γ is essential for transepithelial neutrophil trafficking and polymorphonuclear cell-dependent vascular injury in lung injury induced by LPS [39,40]. The inhibition of PI3K-γ activity through genetic knockout or pharmacological agents can mitigate lung edema in VILI and organ damage in sepsis [41,42]. The relationship between ER stress and PI3K-γ in critical illnesses is not well established. Given that clinical patients with sepsis often require MV life support, researchers should strive to avoid underestimating the deleterious effects of mechanical stretch. In a murine study of traumatic brain injury, Liu et al. reported that the activation of neuronal PI3K-γ promoted ER stress and associated neuronal cell death and long-term functional impairment [43]. These authors demonstrated that inhibition of PI3K-γ attenuated neuronal ER stress and CHOP-mediated neuronal apoptosis and contributed to neuroprotection in acute and chronic traumatic brain injury [43]. Similarly, our results suggest PI3K-γ overexpression was induced by high mechanical lung stretch or synergistic LPS-induced lung inflammation, simulating the real clinical scenario of patients with sepsis receiving MV. LPS and mechanical stress induced by ventilation can activate ROS and inflammatory signaling. These triggered ER stress and associated upstream and downstream UPR pathways, aggravating pulmonary oxidative stress, inflammation, apoptosis, and injury through PI3K-γ. Moreover, inhibition of PI3K-γ through genetic knockout or the ER stress inhibitor 4-PBA reduced injuries associated with ER stress in septic mice receiving MV.

This study has some limitations. First, although we demonstrated the morphological alterations of aberrant and enlarged ER in the injury groups compared to the controls by TEM (Figure 1A–E), the specific identification of ER would be expected to be presented by using the ER markers in future studies [44]. Second, Eganelisib is a promising, highly selective PI3K inhibitor for the clinical purpose of treating cancers and inflammatory diseases [45]. Based on our present study results, this orally administered PI3K-γ inhibitor is anticipated to provide a clinical therapeutic approach to treat ALI in sepsis.

## 4. Materials and Methods

### 4.1. Experimental Animals

We purchased PI3K-γ-deficient or wild-type C57BL/6 mice, aged between 6 and 8 weeks and weighing between 20 and 25 g, from the National Laboratory Animal Center (Taipei, Taiwan) and Jackson Laboratories (catalog number 024587 (PI3K-γ), Bar Harbor, ME, USA) [14]. Concisely, a defective neutrophil chemotaxis and respiratory burst by homozygote mutants (PI3K-γ^−/−^) is displayed as a reaction to formyl peptide N-formyl-Met-Leu-Phe (fMLP) and C5a induction, as well as incitement of mature T lymphocytes and diminished thymocyte survival [14]. PI3K-γ^−/−^ mice were confirmed by the lower measurements of the PI3K-γ protein employing a Western blot analysis. The study was conducted in strict accordance with the Guidance for the Use and Care of Laboratory Animals of the National Institutes of Health. The experimental protocol was ratified by the Institutional Animal Care and Use Committee of Chang Gung Memorial Hospital (Permit number: 2022120505). We performed all surgeries under xylazine and ketamine anesthesia, and we made all efforts to diminish suffering.

### 4.2. Pharmacological Inhibitors

ER stress inhibitor (4-PBA, Sigma, St. Louis, MO, USA) 10 mg/kg was administered intraperitoneally 1 day before MV based on our present and previous studies that demonstrated 10 mg/kg inhibited ER stress activity [15].

### 4.3. Measurement of Inflammatory Cytokines

IL-6 with a lower measurement limit of 1.8 pg/mL and MMP-9 (14 pg/mL) were identified in bronchoalveolar lavage fluid, applying immunoassay kits containing primary polyclonal anti-mouse antibodies that were cross-reactive with rat and mouse IL-6, and MMP-9 (Biosource International, Camarillo, CA, USA). Each sample was detected in duplicate as indicated by the manufacturer’s instructions.

### 4.4. Immunoblot Analysis of ER Stress and PI3K-γ

We homogenized the mouse lungs in 0.5 mL of lysis buffer as previously depicted [4,46]. We matched crude cell lysates for protein concentration, resolved on a 10% bis-acrylamide gel, and electrotransferred to Immobilon-P membranes (Millipore Corp., Bedford, MA, USA). For assessment of ER stress (phospho-IRE1α, phospho-PERK, cleaved ATF6, GRP78, CHOP, caspase 3), PI3K-γ, and GAPDH, we manipulated Western blot analyses with respective antibodies (New England BioLabs, Beverly, MA, USA, Santa Cruz Biotechnology, Santa Cruz, CA, USA, and Novus Biologicals, Littleton, CO, USA). Blots were produced by enhanced chemiluminescence (NEN Life Science Products, Boston, MA, USA).

### 4.5. Statistical Analysis

The Western blots were quantified using a National Institutes of Health (NIH) image analyzer, Image J 1.27z (National Institutes of Health, Bethesda, MD, USA), and exhibited as arbitrary units. Values were presented as the mean ± SD from at least 5 separate experiments. The data of malondialdehyde (MDA), sodium dismutase (SOD), histopathologic assay, and oxygenation were analyzed using Statview 5.0 (Abascus Concepts Inc., Cary, NC, USA; SAS Institute, Inc., Cary, NC, USA). All results of Western blots were normalized to the non-ventilated control wild-type mice with room air. ANOVA was applied to evaluate the statistical significance of the differences, followed by multiple comparisons with a Scheffe’s test, and a *p* value < 0.05 was judged as statistically significant.

Ventilator protocol, lipopolysaccharide administration, analysis of bronchoalveolar lavage fluid total protein, Evans blue dye (EBD) analysis, analysis of lung water, immunohistochemistry of PI3K-γ, measurement of oxidative and antioxidant biomarkers, and TEM were performed as previously depicted [4,46].

## 5. Conclusions

In conclusion, our results illustrate that therapies targeting PI3K-γ can attenuate ER stress signaling, oxidants, inflammation, lung edema, and epithelial apoptosis. PI3K-γ inhibition considerably improved biomolecular, pathological, and functional impairments in our animal model of VILI with sepsis. Further clarification of the synergistic effects of mechanical stretch and endotoxin on PI3K-γ signaling can elucidate the pathogenetic mechanisms of ALI related to MV and sepsis. Considering the lack of effective target therapies currently available for patients with sepsis, pharmacological agents targeting PI3K-γ that attenuate VILI in septic mice may represent a novel therapeutic option for ALI treatment in patients with sepsis receiving MV support.

## Figures and Tables

**Figure 1 ijms-26-05761-f001:**
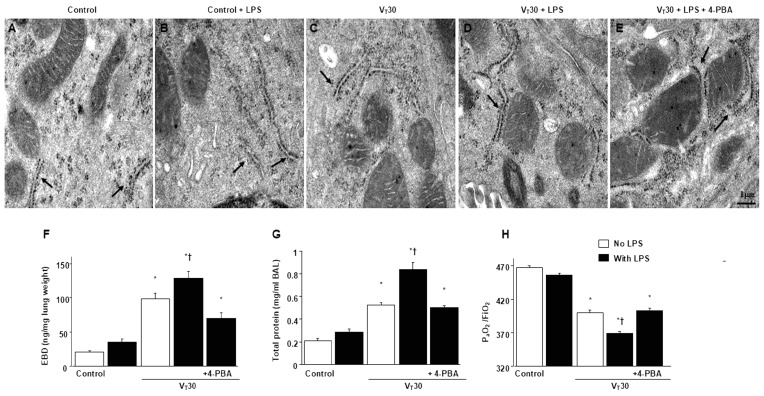
Reduction in endotoxin-enhanced lung stretch-induced lung injury and ultrastructure of endoplasmic reticulum by 4-PBA. (**A**–**E**) Representative micrographs of the longitudinal sections (×160,000) were from the lungs of nonventilated control mice and mice ventilated with a tidal volume (V_T_) of 30 mL/kg for 5 h with or without LPS administration (*n* = 3 per group). Morphologically aberrant and enlarged ER in bronchial epithelium are indicated by arrows. (**F**) Evans blue dye analysis, (**G**) total protein, and (**H**) PaO_2_/FiO_2_ (*n* = 5 per group). 4-PBA 10 mg/kg was given intraperitoneally 1 day before mechanical ventilation. Scale bars represent 1 μm. * *p* < 0.05 versus the nonventilated control mice with LPS; † *p* < 0.05 versus all other groups.

**Figure 2 ijms-26-05761-f002:**
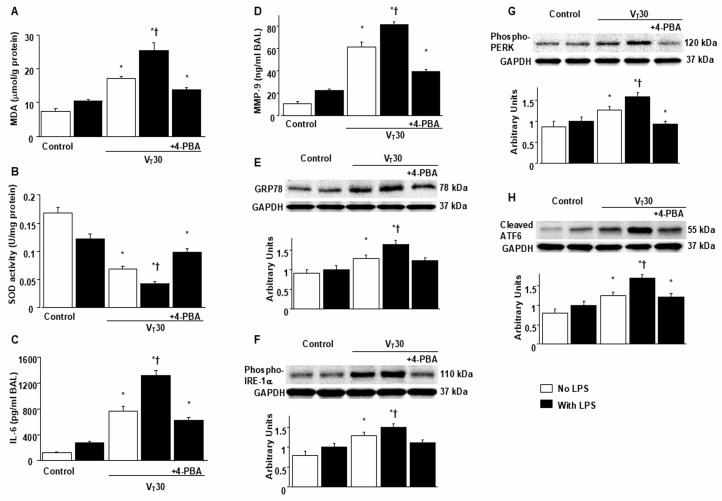
Inhibition of endotoxin-aggravated lung stretch-induced oxidative stress, IL-6 and MMP-9 production, and endoplasmic reticulum stress protein expression by 4-PBA. (**A**) MDA, (**B**) SOD, (**C**) BAL IL-6, and (**D**) BAL MMP-9 were from the lungs of nonventilated control mice and those subjected to a tidal volume of 30 mL/kg for 5 h with or without LPS administration (*n* = 5 per group). Western blots were performed employing antibodies that recognize (**E**) GRP78, (**F**) phosphorylated IRE-1α, (**G**) phosphorylated PERK, (**H**) cleaved ATF6, and GAPDH expression in lung tissues from nonventilated control mice and mice ventilated with a tidal volume of 30 mL/kg for 5 h with or without LPS administration (*n* = 5 per group). Arbitrary units were expressed as the ratio of relative GRP78, phosphorylated IRE-1α, phosphorylated-PERK, and cleaved ATF6 activation (*n* = 5 per group). 4-PBA 10 mg/kg was given intraperitoneally 1 day before mechanical ventilation. * *p* < 0.05 versus the nonventilated control mice with LPS; † *p* < 0.05 versus all other groups.

**Figure 3 ijms-26-05761-f003:**
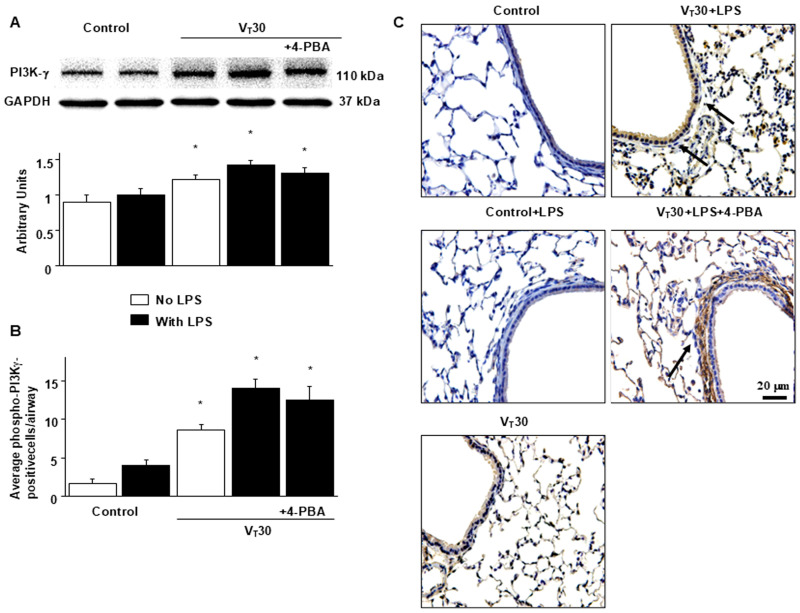
Endotoxin-augmented lung stretch-induced PI3K-γ protein expression. Western blots were performed employing antibodies that recognize (**A**) PI3K-γ and GAPDH expression in lung tissues from nonventilated control mice and mice ventilated with a tidal volume of 30 mL/kg for 5 h with or without LPS administration (*n* = 5 per group). Arbitrary units were expressed as the ratio of relative PI3K-γ activation (*n* = 5 per group). (**B**,**C**) Representative micrographs (×400) with PI3K-γ staining of paraffin lung sections and quantification were obtained from nonventilated control mice and mice ventilated with a tidal volume of 30 mL/kg for 5 h with or without LPS administration (*n* = 5 per group). A dark-brown diaminobenzidine signal indicated by arrows showed positive staining of PI3K-γ in the airway epithelia. 4-PBA 10 mg/kg was given intraperitoneally 1 day before mechanical ventilation. Scale bars represent 20 μm. * *p* < 0.05 versus the nonventilated control mice with LPS.

**Figure 4 ijms-26-05761-f004:**
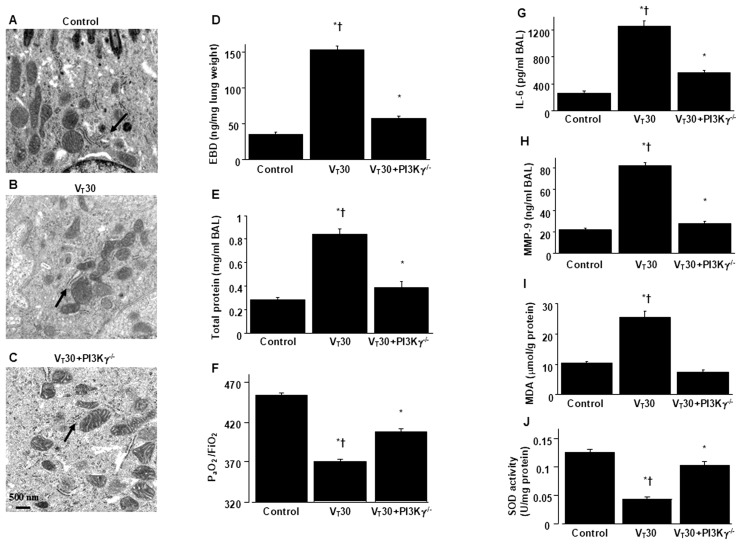
Reduction in lung stretch-induced lung inflammation in PI3K-γ-deficient mice. (**A**–**C**) Representative micrographs of the longitudinal sections (×75,000) were from the lungs of nonventilated control mice and mice ventilated with a tidal volume of 30 mL/kg for 5 h with LPS administration (*n* = 3 per group). Morphologically aberrant and enlarged ER in bronchial epithelium is identified by arrows. (**D**) Evans blue dye analysis, (**E**) total protein, (**F**) PaO_2_/FiO_2_, (**G**) IL-6, (**H**) MMP-9, (**I**) MDA, and (**J**) SOD were from the lungs of nonventilated control mice and those of mice ventilated with a tidal volume of 30 mL/kg for 5 h with LPS administration (*n* = 5 per group). * *p* < 0.05 versus the nonventilated control mice with LPS; † *p* < 0.05 versus PI3K-γ-deficient mice.

**Figure 5 ijms-26-05761-f005:**
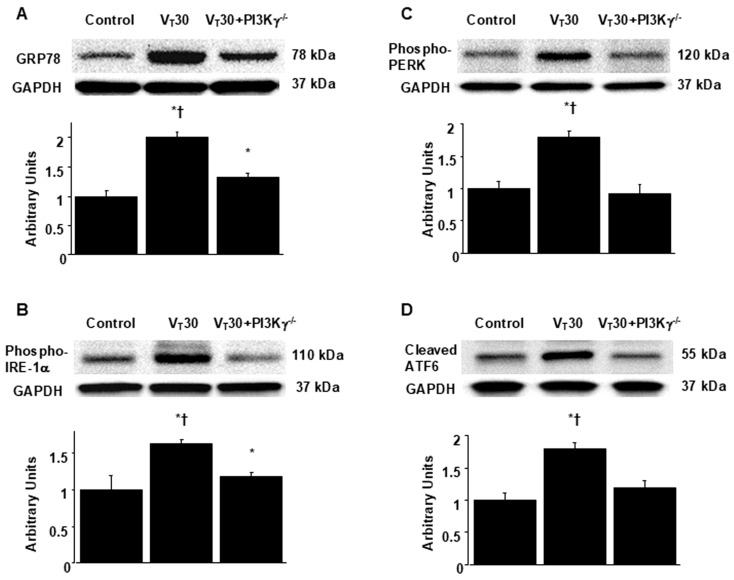
Inhibition of endotoxin-augmented lung stretch-induced endoplasmic reticulum stress in PI3K-γ-deficient mice. Western blots were performed employing antibodies that recognize (**A**) GRP78, (**B**) phosphorylated IRE-1α, (**C**) phosphorylated PERK, (**D**) cleaved ATF6, and GAPDH expression in lung tissues from nonventilated control mice and mice ventilated with a tidal volume of 30 mL/kg for 5 h with LPS administration (*n* = 5 per group). Arbitrary units were expressed as the ratio of relative GRP78, phosphorylated IRE-1α, phosphorylated-PERK, and cleaved ATF6 activation (*n* = 5 per group). * *p* < 0.05 versus the nonventilated control mice with LPS; † *p* < 0.05 versus PI3K-γ-deficient mice.

**Figure 6 ijms-26-05761-f006:**
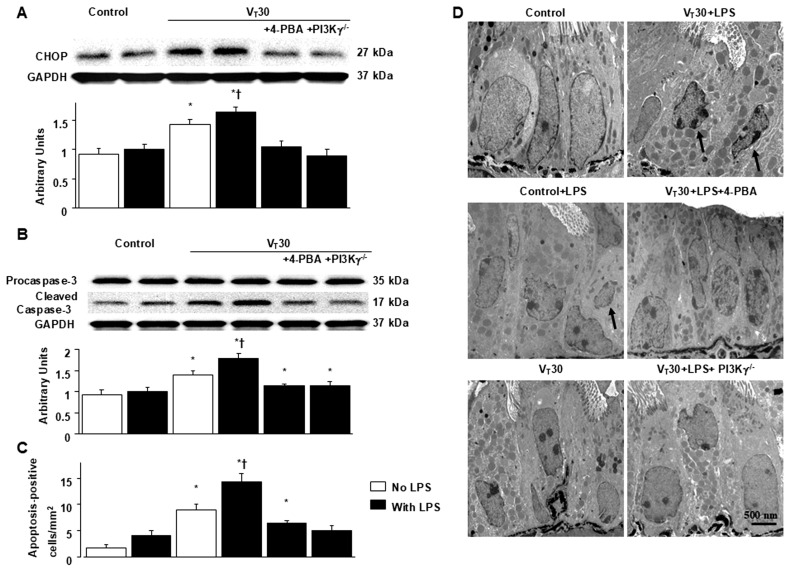
PI3K-γ homozygous knockout ameliorated lung stretch-induced CHOP and caspase-3 expression and epithelial apoptosis. (**A**,**B**) Western blots were conducted employing antibodies that recognize caspase-3, CHOP, and GAPDH expression in lung tissue from nonventilated control mice and mice ventilated with a tidal volume of 30 mL/kg for 5 h with or without LPS administration (*n* = 5 per group). Arbitrary units were expressed as the ratio of cleaved caspase-3 and CHOP to GAPDH (*n* = 5 per group). (**C**,**D**) Representative TUNEL staining of paraffin lung sections (Appendix A) and quantitation (×400, *n* = 5 per group) and TEM micrographs (×11,000, *n* = 3 per group) of the lung sections were from the lungs of nonventilated control mice and mice ventilated with a tidal volume of 30 mL/kg for 5 h with or without LPS administration (*n* = 3 per group). Apoptosis is characterized by highly condensed and fragmented heterochromatin of epithelial cells. Apoptotic cells are indicated by arrows. Scale bars represent 500 nm. * *p* < 0.05 versus the nonventilated control mice with LPS; † *p* < 0.05 versus all other groups.

**Figure 7 ijms-26-05761-f007:**
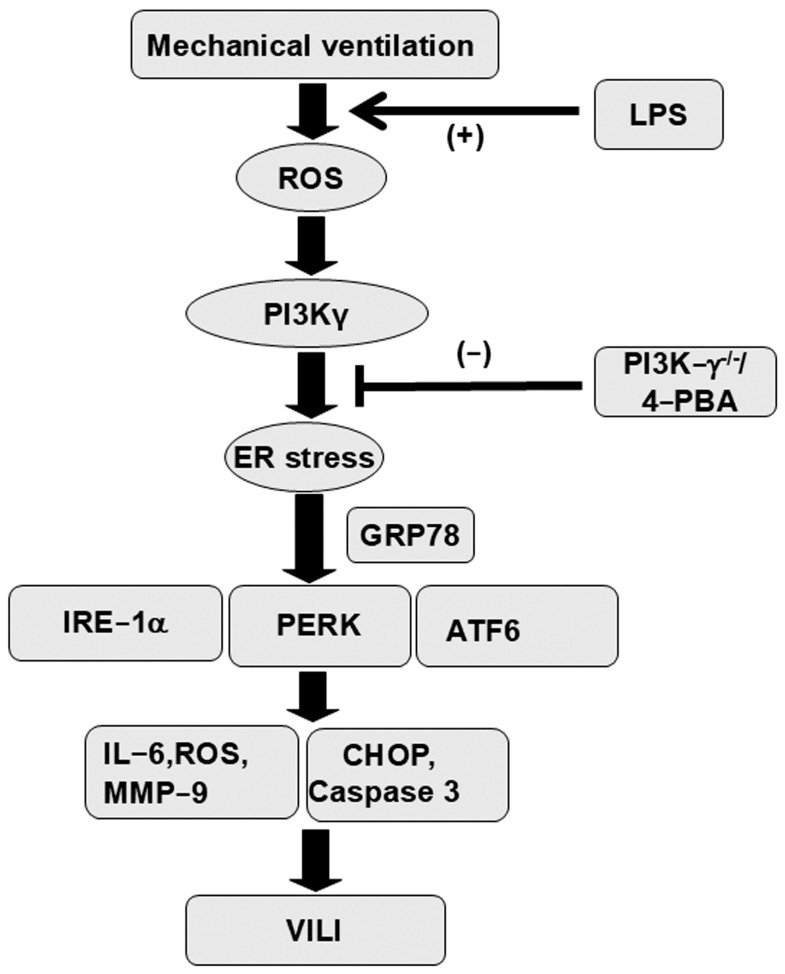
Schematic figure illustrating the signaling pathway activation with mechanical ventilation and endotoxemia. Endotoxin-enhanced augmentation of mechanical stretch-mediated cytokine production, ER stress, and lung injury was reduced by the administration of 4-PBA and with PI3K-γ homozygous knockout.

## Data Availability

The data presented in this study are available on request from the corresponding author.

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
