# Peer review of "Attenuation of Ventilation-Induced Endoplasmic Reticulum Stress Associated with Lung Injury Through Phosphoinositide 3-Kinase-Gamma in a Murine Endotoxemia Model"

_ijms, 2025, doi:10.3390/ijms26125761_

Round 1

Reviewer 1 Report

Comments and Suggestions for Authors

Manuscript ID: ijms-3598788

Attenuation of Ventilation-Induced Endoplasmic Reticulum Stress Associated with Lung Injury through Phosphoinositide 3-Kinase-Gamma in a Murine Endotoxemia Model

Patients with sepsis are often required treatment with mechanical ventilation (MV), and MV sometimes causes ventilation-induced lung injury (VILI). Endoplasmic reticulum (ER) stress is caused by MV, and it is also reported that phosphoinositide-3 kinase (PI3K)-gamma modulates ER stress. In the present study, the authors analyzed the involvement of ER stress and PI3K-gamma in mice receiving MV and LPS administration, and found that MV augmented acute lung injury (ALI) after toxemia partly depending on the PI3K-gamma pathway in mice, suggesting a possibility of therapeutic option for VILI in sepsis.

  Although this manuscript is of some interest, the purpose and rationale of this study are not clearly described as detailed below.

  1. The purpose of the study, experimental design, and conclusions are not clearly described: especially it is not clear whether they fucus on ER stress and MV with sepsis or ER stress and MV without sepsis. The authors compare the effect of MV with and without LPS, but administration of 4-PBA or PI3K-gamma K/O were performed only in MV with LPS mice.
  2. 1A and Fig. 4C are completely the same except the height and width ratio of the photo is different. All the TEM figures should be confirmed if they are correctly represented.
  3. In Figs. 1A-E, explain what the black arrows indicate. The morphological changes in the ER (lines 116-119, 124-126) is not clear. Add enlarged figures showing these aberrant ER.
  4. Related with the comments in 3, in Fig. 1, morphological changes of the ER may be easily detected by immunofluorescent microscopy of the cells using ER markers. Add these results.
  5. In Fig. 2H and Fig. 5D, the authors analyze the upregulation of ATF6 protein as an ER stress marker. ATF6 is cleaved by ER stress, which is the hallmark of UPR. ATF6 cleavage should be analyzed instead.
  6. In Fig. 3C, explain the arrows. Also, explain the strong signal of PI3K-gamma detected under the epithelial cell layer in the lung of mice treated with VT30+LPS+4-PBA.
  7. In Fig. 6C and D, the authors analyze the apoptosis-positive cells by TEM. The calculation of apoptotic cell numbers would be easier and more accurate using cell staining with apoptosis detection kit or others. Explain the reason why they used TEM in this assay.
  8. Part of the discussion is not clearly described.
  9. TEM is missing in Materials and methods.

Minor points:

  1. Typos and grammatical errors should be corrected. E.g., ERAD (line 72), phopho-IRE1alpha (line 135), and others.
  2. Remove abbreviations in each figure legend.

Author Response

Reply to Reviewer 1 

Reviewer 2 Report

Comments and Suggestions for Authors

A murine model of VILI, including high VT (30 ml/kg) and LPS, was used to delineate the mechanistic role of PI3Kγ. A PI3Kγ KO mouse strain and 4-PBA were used as probes to validate the findings. The involvement of ER stress and the PI3Kγ signaling pathway in VILI was reported 15 years ago. The connection between PI3Kγ and ER stress has also been well-described in bleomycin-induced lung injury. This manuscript attempts to link ER stress and PI3Kγ signaling together.  Interestingly, the specific PI3K inhibitor, IPI-549 (Eganelisib), was not used to validate your findings, as it may have therapeutic potential.

               The introduction provides lengthy detail about the PI3Kγ and ER stress, or UPS, including the downstream XBP-1 splicing and enhanced ATF4 expression. However, these were not even explored in this manuscript. Caspase-12 is considered the ER-specific apoptosis signaling, and its cleaved form is upstream of the caspase-3 cleavage. I am curious why caspase-12 was not studied? Characteristic ER structural changes under ER stress include increased ER caliber and increased distance between the ER and mitochondria. Providing a TEM image of secretory cells or type 2 alveolar cells with more ER structure will be better. I do not understand why bronchial epithelium was used. The ER structure pointed out in Figure 1E is not convincing. TEM might not be the best way to quantify apoptosis, as only a limited number of nuclei can be analyzed. Most nuclei in Figure 6D, except the two for control, will be counted at different stages of apoptosis with some extent of condensation and fragmentation. I am curious why you did not use other image methods to quantify apoptosis.   

I would call Grp78/HspA5/BiP an endogenous chaperone instead of a transducer in the ER. The increased Grp78 expression is a biomarker for ER stress but does not generate incorrect protein structure. The increased Grp78 attenuates the UPR, a survival mechanism, unless it persists under prolonged exogenous stress. ATF6 is cleaved by the Golgi site-1 proteases after leaving the Grp78 under ER stress. The cleaved ATF6 (~55 kDa) instead of the full-length (~100 kDa) ATF6 is considered the biomarker of ER stress.

I want to suggest the immunoblot expression analysis in the same format. In Figure 5, controls were treated as one while the same adjustment was not applied in Figures 2, 3, and 6.

Minor concerns

Lines 386-388: “After 1 h of spontaneous respiration to allow for development of a septic response, the mouse will be subjected to MV for 5 h [4,44].” I question the temporal relationship you described here and the tense of the sentence.

Lines 403-404: “For assay of ER stress (PI3Kγ, phosphor-IRE1, phosphor-PERK, ATF6, GRP78, CHOP, caspase 3),…” I do not consider PI3Kγ as a direct member of ER stress.

Comments on the Quality of English Language

There are minor grammatical issues, but I have no difficulty reading through them.

Author Response

See attached files

Round 2

Reviewer 1 Report

Comments and Suggestions for Authors

Author Response

Attenuation of Ventilation-Induced Endoplasmic Reticulum Stress Associated with Lung Injury through Phosphoinositide 3-Kinase-Gamma in a Murine Endotoxemia Model

  1. In Figs. 1A-E, explain what the black arrows indicate. The morphological changes in the ER (lines 116-119, 124-126) is not clear. Add enlarged figures showing these aberrant ER.

Answer: Thanks for the reviewer’s opinions. The black arrows indicate the morphologically aberrant and enlarged ER structures. Because of the technical factor for our TEM, we have added the description in our Discussion part (Line 376-380) as “This study has some limitations. First, although we demonstrated the morphological alterations of aberrant and enlarged ER in the injury groups compared to the controls by TEM (Figure 1A–E), the highly precise images or specific identification of ER and adjacent mitochondria would be expected to be presented by promoting our analysis techniques for ultrastructural pathology or using the ER markers in the future studies [44].”

At least, it gives some additional information if the authors show enlarged figures of the aberrant ER in insets. In control (Fig. 1a), there is a black arrow indicating the existence of aberrant ER, thus, it is not clear if there is a difference between the control and other experimental groups. It is necessary to show quantitative data of the aberrant ER.

Answer: Thanks for the reviewer’s suggestions. We have replaced the Fig. 1A to 1E by new clearer Figures with magnification to x160,000 as best as we can do. It makes clearer to show the differences between the control and other experimental groups. The experiment of immunofluorescent staining using the ER markers is ongoing, and we will add these results when the data are completed.

  1. Related with the comments in 3, in Fig. 1, morphological changes of the ER may be easily detected by immunofluorescent microscopy of the cells using ER markers. Add these results.

Answer: Thanks for the reviewer’s comments. Because of the technical factor for our TEM, we have added the description in our Discussion part (Line 376-380) as “This study has some limitations. First, although we demonstrated the morphological alterations of aberrant and enlarged ER in the injury groups compared to the controls by TEM (Figure 1A–E), the highly precise images or specific identification of ER and adjacent mitochondria would be expected to be presented by promoting our analysis techniques for ultrastructural pathology or using the ER markers in the future studies [44].”

Immunofluorescent microscopy of the cells is not a difficult technique. These data should be added. This will partly answer to the criticism in 3.

Answer: Thanks for the reviewer’s suggestions. The experiment of immunofluorescent staining using the ER makers is ongoing, and we will add these results when the data are completed.

  1. In Fig. 2H and Fig. 5D, the authors analyze the upregulation of ATF6 protein as an ER stress marker. ATF6 is cleaved by ER stress, which is the hallmark of UPR. ATF6 cleavage should be analyzed instead.

Answer: Thanks for the reviewer’s opinions. We have added the descriptions in our Discussion part (Line 380-386) as “Second, we demonstrated the upregulation of ATF6 protein in the injury groups as an ER stress sensor, however ATF6 is translocated to the Golgi apparatus and cleaved into its active form [25]. The active form of ATF6 is found to be rapidly degraded and difficult to measure by western blotting unless using proteasome inhibitors [45]. To analyze the active form of ATF6 to improve the evaluation of ER stress might warrant further consideration for future work.”

Please show the cleaved form of ATF6. Detection of cleaved form of ATF6 without proteasome inhibitors is reported in:

https://doi.org/10.1091/mbc.10.11.3787

DOI 10.1016/j.devcel.2007.07.018

Answer: Thanks for the reviewer’s comments. The experiment of analyzing cleaved ATF6 is ongoing, and we will add these results when the data are completed.

  1. In Fig. 3C, explain the arrows. Also, explain the strong signal of PI3K-gamma detected under the epithelial cell layer in the lung of mice treated with VT30+LPS+4-PBA.

Answer: Thanks for the reviewer’s comments. We have added the explanation of the arrows in Line 200-201 as “A dark-brown diaminobenzidine signal indicated by arrows showed positive staining of PI3K-γ in the airway epithelia.” Further, the reason why the strong signal of PI3K-γ was detected under the epithelial cell layer in the lung of mice treated with VT30+LPS+4-PBA in our Fig. 3C is described in Line 179-181 as below: “No decrease in PI3K-γ expression after 4-PBA administration was observed, indicating that ER stress markers are downstream in the PI3K-γ-induced signaling cascade.”

To answer to “explain the strong signal of PI3K-gamma detected under the epithelial cell layer in the lung of mice treated with VT30+LPS+4-PBA”, please explain the specificity of the localization (under the epithelial cell layer) of the signal in this condition.

Answer: ALI is characterized by impaired gas exchange of alveoli and airways, which includes terminal and respiratory bronchioles. Therefore, we explored high-tidal-volume MV and LPS-induced lung injury in the murine bronchial epithelium like our previous study in mice and other study in human [1, 2]. In the current study, bronchial epithelial cells were our main focus to investigate. Immunohistochemistry was used to further define the cell types involved in the ALI-induced PI3K-γ expression.

[1]. Li LF, et al. Attenuation of Ventilation-Enhanced Epithelial-Mesenchymal Transition through the Phosphoinositide 3-Kinase-γ in a Murine Bleomycin-Induced Acute Lung Injury Model. Int J Mol Sci. 2023;24(6):5538.

[2]. Jiang R, et al. Aspirin Inhibits LPS-Induced Expression of PI3K/Akt, ERK, NF-κB, CX3CL1, and MMPs in Human Bronchial Epithelial Cells. Inflammation. 2016;39(2):643-50.

  1. Part of the discussion is not clearly described.

Answer: We have revised the discussion with our study limitations to make the contents clearly to be understood by our readers.

Lines 346-348, 4): not correctly described.

Answer: We have made revision as in Line 301-303: ”4) reduce overexpression of the ER stress response protein GRP78 and ER transmembrane sensors PERK, IRE1a, and ATF6;”.

Thanks for the reviewer’s comments. The GRP78 is not the ER stress activator and is known as the ER stress response protein to cope with ER stress.

Lines 358-9: not accurate

Answer: We have made revision as in Line 313-314: “The ER, a vital intracellular organelle, facilitates protein folding, translocation, and posttranslational modifications in the secretory pathways [24].”.

Lines 361-3: not correct.

Answer: We have made revision as in Line 315-318: “During ER stress, UPR is produced to recover ER function. When ER stress overwhelms the UPR, it triggers apoptosis, causing cellular injury and death [24]”.

Lines 419-421, with Fig. 7: the relationship of PI3K-gamma and ROS is not clear.

Answer: We have made revision as in Line 66-72: “ER stress induced by pathological conditions, such as MV, oxidative stress, and sepsis, causes GRP78 to dissociate from transmembrane proteins, which are then activated to sense ER stress and initiate the UPR signaling [6,10,11]. GRP78 stimulates protein modification, oligomerization, and refolding in the ER for structural correction. However, the protein refolding produces reactive oxygen species (ROS) that aggravate oxidative stress.”. In addition, we have made revision as in Line 376-381:” LPS and mechanical stress induced by ventilation can activate ROS and inflammatory signaling. These triggered ER stress and associated upstream and downstream UPR pathways, aggravating pulmonary oxidative stress, inflammation, apoptosis, and injury through PI3K-γ. Moreover, inhibition of PI3K-γ through genetic knockout or the ER stress inhibitor 4-PBA reduced injuries associated with ER stress in septic mice receiving MV.”. Our Fig. 7 illustrated that ROS produce a vicious cycle of acute lung inflammation and injury through PI3K-γ and ER stress in the present study.

Minor points:

  1. Typos and grammatical errors should be corrected. E.g., ERAD (line 72), phopho-IRE1alpha (line 135), and others.

Answer: We have revised the Typos and grammatical errors as we can do.

The editorial staff of the journal will confirm the typos and grammatical errors.

Answer: Thanks for the reviewer’s suggestions.

Reviewer 2 Report

Comments and Suggestions for Authors

I still have concerns about the TEMs in Figure 1. Increased distance between ER and mitochondria is a typical finding under UPR due to impaired MAM formation, and increased ER caliber is another common finding. The images you provided are not impressive. Non-secretory bronchial epithelial cells usually do not contain much ER structure like AT2 cells. The image of the control epithelial cell has a small amount of ER structure, and the ER pointed out by the arrow is far away from the mitochondria. The label (A-E) for each micrograph in Figure 1 should be added. Lack of funding is probably not a good response to the reviewer’s comment.

I agree that CHOP initiates apoptosis under UPR with caspase-3 cleavage as the final step. However, caspase-12 cleavage, upstream of caspase-3 cleavage, is a typical biomarker of UPR-induced apoptosis in rodents (mice and rats). Although the existence of a human caspase-12 homologue has been questioned, recent research suggests its existence in humans. I have no difficulty identifying cleaved ATF6 (~55 kDa) by immunoblots in both rat and mouse lungs.

Author Response

I still have concerns about the TEMs in Figure 1. Increased distance between ER and mitochondria is a typical finding under UPR due to impaired MAM formation, and increased ER caliber is another common finding. The images you provided are not impressive. Non-secretory bronchial epithelial cells usually do not contain much ER structure like AT2 cells. The image of the control epithelial cell has a small amount of ER structure, and the ER pointed out by the arrow is far away from the mitochondria. The label (A-E) for each micrograph in Figure 1 should be added. Lack of funding is probably not a good response to the reviewer’s comment.

Answer: Thanks for the reviewer’s suggestions. We have replaced the Fig. 1A to 1E by new clearer Figures with magnification to x160,000 as best as we can do. It makes clearer to show the differences between the control and other experimental groups.

I agree that CHOP initiates apoptosis under UPR with caspase-3 cleavage as the final step. However, caspase-12 cleavage, upstream of caspase-3 cleavage, is a typical biomarker of UPR-induced apoptosis in rodents (mice and rats). Although the existence of a human caspase-12 homologue has been questioned, recent research suggests its existence in humans. I have no difficulty identifying cleaved ATF6 (~55 kDa) by immunoblots in both rat and mouse lungs.

Answer: Thanks for the reviewer’s comments. The experiment of analyzing cleaved ATF6 is ongoing, and we will add these results when the data are completed.

Round 3

Reviewer 1 Report

Comments and Suggestions for Authors

Look forward to seeing the additional ongoing results.

Author Response

Comments and Suggestions for Authors from the Reviewer 1

Look forward to seeing the additional ongoing results.

Answer: Thanks for the reviewer’s comments. We have added our additional ongoing results into the Figure 2H and Figure 5D. We have replaced the ATF6 western blot analysis by the cleaved ATF6 western blot analysis in our revised manuscript.